# Early Diagnosis of Neurodegenerative Diseases: What Has Been Undertaken to Promote the Transition from PET to Fluorescence Tracers

**DOI:** 10.3390/molecules29030722

**Published:** 2024-02-04

**Authors:** Nicolò Bisi, Luca Pinzi, Giulio Rastelli, Nicolò Tonali

**Affiliations:** 1Université Paris-Saclay, CNRS, BioCIS, Bat. Henri Moissan, 17, Av. des Sciences, 91400 Orsay, France; 2Department of Life Sciences, University of Modena and Reggio Emilia, Via Giuseppe Campi 103, 41125 Modena, Italy; luca.pinzi@unimore.it (L.P.); giulio.rastelli@unimore.it (G.R.)

**Keywords:** Aβ_1–42_, Tau, α-synuclein, probes, neurodegeneration, Alzheimer’s disease, Parkinson’s disease, tauopathies

## Abstract

Alzheimer’s Disease (AD) and Parkinson’s Disease (PD) represent two among the most frequent neurodegenerative diseases worldwide. A common hallmark of these pathologies is the misfolding and consequent aggregation of amyloid proteins into soluble oligomers and insoluble β-sheet-rich fibrils, which ultimately lead to neurotoxicity and cell death. After a hundred years of research on the subject, this is the only reliable histopathological feature in our hands. Since AD and PD are diagnosed only once neuronal death and the first symptoms have appeared, the early detection of these diseases is currently impossible. At present, there is no effective drug available, and patients are left with symptomatic and inconclusive therapies. Several reasons could be associated with the lack of effective therapeutic treatments. One of the most important factors is the lack of selective probes capable of detecting, as early as possible, the most toxic amyloid species involved in the onset of these pathologies. In this regard, chemical probes able to detect and distinguish among different amyloid aggregates are urgently needed. In this article, we will review and put into perspective results from ex vivo and in vivo studies performed on compounds specifically interacting with such early species. Following a general overview on the three different amyloid proteins leading to insoluble β-sheet-rich amyloid deposits (amyloid β_1–42_ peptide, Tau, and α-synuclein), a list of the advantages and disadvantages of the approaches employed to date is discussed, with particular attention paid to the translation of fluorescence imaging into clinical applications. Furthermore, we also discuss how the progress achieved in detecting the amyloids of one neurodegenerative disease could be leveraged for research into another amyloidosis. As evidenced by a critical analysis of the state of the art, substantial work still needs to be conducted. Indeed, the early diagnosis of neurodegenerative diseases is a priority, and we believe that this review could be a useful tool for better investigating this field.

## 1. Introduction

Neurodegenerative diseases represent one of the main causes of public health concerns to date, affecting almost 179 million people worldwide and costing more than EUR 800 billion only in Europe [1]. Alzheimer’s Disease (AD) and Parkinson’s Disease (PD) are the first and the second most common neurodegenerative diseases, respectively, both of them being amyloidopathies in which an amyloid protein misfolds and aggregates, causing neurotoxicity and cell death [2]. In the case of AD, the amyloids involved are tubulin associated unit (Tau) and Aβ_1–42_, while in PD, α-synuclein (αSyn) is the one whose misfolding and aggregation leads to toxic inclusions and neuronal death. Throughout the preceding century, researchers endeavored to elucidate the primary pathways and pathological features underlying the initiation and progression of AD and PD; although many hypotheses have been put forward, the only histopathological feature characterizing these diseases is still represented by amyloid inclusions. In particular, these entities are Lewy Bodies (LB) in PD, and intracellular neurofibrillary tangles (NFT) and extracellular amyloid plaques in AD [3,4]. These histopathological hallmarks strictly correlate with the amyloid hypothesis, which represents the most studied, but also the most controversial one. The association of this hypothesis with the aforementioned histopathological features underscores its prominence as one of the primary hypotheses of neurodegeneration [4].

Nowadays, AD diagnosis is based on a clinical evaluation and imaging investigation based on techniques such as positron emission tomography (PET), while a definitive diagnosis is confirmed only upon a post mortem examination of the patients’ brain. The diagnosis requires the detection of dopaminergic neuron loss, together with the presence of LB and Lewy neurites for PD. NFT and amyloid plaques are instead required to validate the diagnosis of AD [5]. The diagnostic criteria and methods for other neurodegenerative diseases are even less reliable. CT (Computer Tomography) and MRI (Magnetic Resonance Imaging) scans of patients’ brains are employed to provide information about the shape, position, or volume of the tissue, thus offering an overview of the progress of central nervous system (CNS) tissue deterioration when the disease is at an advanced stage. Several molecular imaging compounds have been studied so far, with four of them being approved for clinical use. In particular, Florbetaben, Florbetapir, and Flutemetamol have been approved for the detection of beta-amyloid plaques in the brain, and Flortaucipir F18 for the detection of Tau neurofibrillary tangles [6,7]. Even though amyloid plaques in the brain are a characteristic feature of AD, their detection through PET imaging cannot be used to diagnose the disease. Indeed, the presence of Tau neurofibrillary tangles correlates better with cognitive symptoms in AD with respect to that of amyloid plaques. Moreover, these latter aggregates are not easily detectable with Aβ PET tracers. In addition, Aβ aggregates cannot be considered as a specific hallmark of AD, as amyloid plaques are frequently also found in dementia with Lewy bodies (i.e., the second most common degenerative dementia), as well as in blood vessels in cerebral amyloid angiopathy. Therefore, patients with these conditions show high signals on amyloid PET scans that are similar in pattern to those seen in AD [8,9].

## 2. Strategies for the Diagnosis of Pre-Symptomatic Neurodegenerative Diseases

Significant advances have been made in amyloidosis imaging so far; however, methods that can help to diagnose and differentiate among patients with neurodegenerative disorders, ideally pre-symptomatically, are still missing. The identification of novel strategies for diagnosis at the incipient stages of Alzheimer’s disease, i.e., before irreversible brain damage or mental decline has emerged, represents one of the most active research areas. Notably, research and clinical findings have highlighted features and biomarkers whose levels significantly change before the onset of early symptoms of these diseases. For example, amyloid beta peptides (Aβ), truncated Tau proteins, and phosphorylated forms of Tau (p-Tau) are few among the main pathological biomarkers whose detection has been progressively implemented, allowing for the detection of a prodromal form of the disease. Their quantification is commonly performed on cerebrospinal fluid (CSF), a medium collected through lumbar punctures. There are three main ELISA-based methods that have been approved as in vitro diagnostic kits for the quantification of Aβ peptides: Innotest^®^ ELISA, IBL International^®^ ELISA, and Euroimmun^®^ ELISA. Despite their good correlation with PET imaging, these methods still present pre-analytical issues, such as: (i) the absorption to the fluid collection tubes, generating false positive results; (ii) the pretreatment of CSF samples using denaturation in guanidine hydrochloride, and (iii) the time and volume of fluid collection not being fully standardized. In addition, even if immunoassay platforms making use of fluorescence, chemiluminescence, or electrochemiluminescence for detection are valuable approaches for quantification (due to their high sensitivity), they still have the disadvantages of inter- and intra-assay variability. This is mainly because of peptide detectability issues that can derive by the high propensity of amyloids to bind to other proteins, often hiding the epitopes recognised by the antibodies. Another drawback of ELISA-based approaches is their intrinsic inability to fully identify pathological oligomers, which may also be undetectable due to their incorporation into larger aggregates. Indeed, antibodies often detect only specific conformations of aberrant proteins; thus, structurally different Aβ species in the sample remain undetected [8,10]. α-synuclein pathological species can be detected in CSF using real-time quaked induced-conversion (RT-QuIC) and protein-misfolding cyclic amplification (PMCA). Interestingly, αSyn pathological aggregates detected by these methods are proven to discriminate between PD and other synucleinopathies such as multiple system atrophy (MSA), suggesting that different polymorphs and strains are present in these diseases [11,12]. In the last years, Tau aggregates have been observed in aberrant quantity within different fluids, the most studied being CSF [13,14,15]. Notably, Tau oligomers aberrantly accumulate in the early phases of tauopathies, and their concentration reflects neurodegeneration progression [13,14,15], particularly for AD [16,17]. While the concentration of biomarkers such as total Tau (t-Tau) and phosphorylated Tau (p-Tau) have been observed to increase early in tauopathies, the diagnosis and discrimination of these pathologies from CSF fluid biomarking present several challenges, such as the invasiveness of fluids collection and the variability across different cohorts of patients [13,16,18,19]. Another potential biofluid for the early detection of tauopathies is blood, whose analysis presents advantages in terms of the personnel and facilities required, being less invasive than CSF collection [20,21,22]. Unfortunately, there are also several issues related to the identification of Tau biomarkers in blood, like the significantly lower concentration of Tau species in this medium compared to CSF [15]. The introduction of ultrasensitive single molecule array (Simoa) technology and mesoscale discovery (MSD) ELISA methods recently enabled the identification of blood Tau biomarkers in large populations of patients [23,24,25], demonstrating that p-Tau may help to discriminate AD from other tauopathies [11,23,24,26,27,28]. However, it should be pointed out that most research related to the identification of Tau biomarkers in blood has been focused on AD [23,26,27,28,29]. Therefore, further efforts are needed to fully understand the utility of blood Tau biomarkers for the early diagnosis and characterization of other tauopathies [30]. Together with blood, urine has also become one of the most studied fluids for Tau biomarkers’ detection [31,32]. For example, it has been recently demonstrated that t-Tau can be detected in urine samples from patients with acute neuronal and glial damage [33]. Considering the low invasiveness and applicability of urine tests in routine controls, future efforts should be addressed towards the identification of biomarkers facilitating the monitoring of the development and progression of tauopathies. Additionally, saliva could also potentially be investigated for pathological Tau detection [34,35,36,37,38,39]. For example, altered p-Tau/T-Tau ratios have been observed in saliva samples from patients with AD, with respect to the control population in a recent study [34]. Similar results were observed by Marksteiner and co-workers [35], who found significantly higher levels of p-Tau181 and lower t-Tau levels in salivary samples from AD patients [35]. On the contrary, similar t-Tau concentrations were observed in saliva from patients with AD, MCI, or healthy controls in a recent study by Ashton et al. [40]. These different outcomes may be explained by saliva composition heterogenicity, as well as restricted patient cohort analysis [41]. Interestingly, multiple immunoassay platforms and cross-sectional studies have found that t-Tau and p-Tau levels vary in the tears of patients affected by neurodegeneration, with respect to healthy controls [42,43,44]. As for saliva, further research in larger cohorts of patients is needed to confirm the predictive value of Tau biomarking in tears for the diagnosis of AD and other tauopathies. Finally, nasal secretions are currently under investigation for the possible detection of tauopathies biomarkers [45,46,47]. Statistically different p-Tau/t-Tau ratios in AD patients were found with respect to the control population in Arrozi’s study, but further research in larger cohorts of patients is needed to confirm the validity of this biofluid for tauopathies monitoring [48]. To date, the detection of Tau aggregates and tauopathies biomarkers in biofluids has gained interest, especially for cost-effective and non-invasive mediums. However, larger patient cohorts are needed, and variability problems still have to be addressed for the efficient early diagnosis of Tau-related diseases.

The use of fluorescent probes represents a valuable methodology for monitoring, in real time, the full-time course of aberrant protein aggregation, from monomeric peptide or protein to amyloid, having single-particle sensitivity. However, the expensiveness of the facilities needed for the analyses and the risk of radiation exposure for the current modalities of molecular imaging applied in clinical studies represent inevitable limitations. Therefore, Near-Infrared (NIR) fluorescent imaging (NIRF) has recently attracted attention as a promising non-invasive method for visualizing amyloid plaques in vivo (eye scan technology) or in biofluids (CSF, urine, and saliva) (Figure 1) [8]. Compared to PET and single-photon emission computed tomography (SPECT), NIRF has many advantages, such as safe and sensitive detection without radiation damage, moderate cost, and minimal auto fluorescence of the NIR probes from cellular and tissue components. Fluorescent amyloid-binding agents offer substantial opportunities for basic research on amyloid composition. Several efforts have been made in the field of bioimaging using non-invasive NIR probes in the frame of neurodegenerative diseases, particularly for AD diagnosis. Despite their favorable features for in vivo application, their translation into preclinical and clinical practices remains challenging, and further optical improvements and technological evolutions are still needed. In particular, there is a need for new NIR fluorescent probes based on new scaffolds, which are able not only to be selective for distinct amyloid aggregates (e.g., Aβ, Tau, hIAPP, or α-synuclein), but also to be selective over soluble species, such as oligomers. The exploration of new probes that selectively target oligomers of one type of amyloid protein is a priority for future research, because oligomers are produced before the accumulation of plaques and thus can be exploited as early biomarkers of the pathological process, long before symptoms have appeared. When new NIR fluorescent probes that are selective for oligomers and able to distinguish different amyloid aggregates are available, it will be possible to translate NIRF imaging into future clinical applications (Figure 1).

In this review, we aim to provide a general overview of the main achievements in the development of fluorescent probes for the detection of amyloid aggregates in the frame of AD and PD. Since a significant number of reports have already been published for the Aβ_1–42_ peptide [49,50,51], this review will firstly show the main advantages and disadvantages of the approaches currently reported for the design of fluorescent probes detecting Aβ_1–42_ toxic oligomers. Afterwards, a discussion in the same regard will be dedicated to the Tau protein, whose detection is still under research investigation, especially for the validation of biomarkers’ outcomes. Finally, a section will be dedicated to α-synuclein, to give the opportunity to the readers interested in fluorescent chemical tools to realize what has been achieved in PD diagnosis to date, and to compare it to that made in AD diagnosis, thus giving the opportunity to take inspiration for future applications in diagnosis. In particular, we focused on three main goals, i.e., (i) to compare and find similarities/differences between the probes developed for αSyn and those concerning Tau and Aβ_1–42_; (ii) to give insights on the use in ex vivo and in vivo systems of these compounds for future diagnosis application; and (iii) to discuss on the main achievements on the detection of amyloids in biofluids. By critically dissecting the strengths and weaknesses of the main probes provided in the literature, we believe this will help the research on neuropathology to advance into neurodegeneration prevention and future early diagnosis without the need for hazardous and costly approaches.

## 3. Approaches in Aβ_1–42_, Tau and αSyn Probing

### 3.1. Aβ_1–42_ Peptide Probes

To date, chemically different NIR probes have been designed and evaluated for binding to various Aβ peptides, especially the insoluble fibers of amyloid plaques (Table 1). Fluorescent probes derivatized from styryl scaffold (Figure 2A) have been proposed [9,52,53,54,55], but even if they were able to cross the blood–brain barrier (BBB), they could not be employed for in vivo imaging because of their low affinity for Aβ plaques and their excitation and emission still outside of the NIR region. Oxazine dyes (Figure 2B) have allowed for improving the fluorescence properties of NIR probes by increasing the quantum yield (41%) upon binding to Aβs and the wavelengths of absorption/emission (650–670 nm) [56].

However, the affinity for Aβ plaques remains moderate and the detection sensitivity is still low because of their small Stokes shift. A series of 2,2′-bithiophene compounds (Figure 2C) possessing the classical push–pull architecture (electron–donor and electron–acceptor groups as terminal moieties, interconnected by a highly polarizable bridge) were reported in the early exploration stage and showed relatively simple structures and excellent fluorescent features, such as a high QY and high emission wavelength (720 nm max) [57,58]. The small planar structure, matching the features of amyloid fibrils surfaces, is responsible of their high binding selectivity to aggregate amyloids, but, at the same time, it is responsible for a lack of specificity for amyloid deposits and high-affinity binding with plasmatic proteins. Inspired by the natural compounds having high binding for plaques, such as curcumin, researchers developed novel NIRF probes with good optical properties: fluorescence intensity increasing upon binding, blue shift, and a large increase in QY (Figure 2D) [59,60,61]. Despite their high affinity for Aβ aggregates and high metabolic stability, these new probes exhibited a low selectivity between Aβ subspecies, making them unsuitable for monitoring Aβ oligomers at a presymptomatic stage of AD. New natural scaffolds for Aβ imaging agents have been exploited after the observation of their direct interaction with Aβ aggregates. Chalcone derivatives (Figure 2E) have been reported as PET/SPECT probes for in vivo imaging and proved to specifically stain the Aβ plaques in brain sections from a transgenic AD model mouse. Starting from these results, a series of chalcone derivatives were developed as NIRF probes with improved characteristics, such as plaque affinity and fluorescent properties [62,63]. Despite this, their low micromolar affinity and short excitation/emission wavelength (400 nm/532 nm) prevent their application. Recently, 1,4-napthoquinones (Figure 2F) have been presented as a novel scaffold for the future designs of drugs and new diagnostic tools that can target both dense-core and diffuse plaques (amorphous deposits that lack dense cores or dystrophic neuritis) [64]. Boron dipyrromethene (BODIPY, Figure 2G) is one of the most widely used small-molecule organic fluorophores in bioimaging. In the literature, several probes based on BODIPY have been proposed for Aβ imaging, but most of them have not yet been reported to image Aβ in vivo due to a high background signal (nonspecific binding) and the difficulty in obtaining a good balance between the polarity of compounds and desirable emission properties [65,66,67,68]. Recent efforts toward the development of new BODIPY-based probes have led to a new PIET (photoinduced electron transfer) quenched NIR probe, containing BODIPY as a fluorophore and tetrahydroquinoxaline as a quenching group. This molecule was found to be able to detect both fibrils and oligomers with significant fluorescent switch-on after binding to soluble and insoluble Aβ species. This new quenching strategy allowed for reducing the intrinsic fluorescence of the probe and thus increasing its QY (quantum yield) upon binding [69].

Finally, the donor–acceptor architecture bridged by a conjugated π-electron chain (push–pull architecture) is still now the method of choice for the design of donor–acceptor NIR probes (DANIR, Figure 2H). The strategy for creating larger conjugated systems is based on the hypothesis that this type of molecules could have more potential to bind to Aβ aggregates and plaques. At the same time, this approach has a bad impact on the QY of the probe, because the unbound probe is already highly fluorescent. Therefore, the research in this field is still working on finding a correct balance between the π-conjugation system and the properties of probes [70,71,72,73]. If the exploitation of diagnostic probes for Aβ fibers and plaques has encountered great research interest in recent years, this is not the case for NIR probes for oligomers binding. A first example is a curcumin-based NIRF imaging probe, which resulted in being unsuitable for in vivo imaging due to its short excitation and emission wavelengths [59]. Different modifications were designed to have a longer π-conjugation system while preserving the binding affinity [74,75]. Until now, CRANAD-3 (curcumin scaffold in Figure 2D) is the probe that exhibited the strongest affinity with Aβ monomers, dimers, and oligomers [76]. However, these curcumin analogs possess a low QY and low selectivity between Aβ subspecies. Specificity was almost achieved after several optimizations of the BODIPY scaffold, which led to the discovery of the BD-Oligo probe. This probe has a high fluorescence enhancement upon incubation with Aβ oligomers, which decreases as more Aβ assembles into fibrils. However, despite its oligomer-sensing ability, this probe suffers from a low binding affinity and short wavelength excitation [77].

The triazole-containing BODIPY-6 and aza-BODIPY are fluorescent dyes showing interactions with soluble and insoluble amyloid aggregates. They have been employed for the co-staining of Aβ in brain tissues and proved to be able to induce a contrasting signal, which can help to monitor the conformational transition of fibrils and oligomers [68,78]. A novel “V-shaped” NIR Aβ oligomer-specific fluorescent probe, PTO-29 [79], demonstrated good photophysical properties and selective recognition of Aβ oligomers over other Aβs in a solution test and phantom imaging study. PTO-29 also showed good BBB penetration and a low cytotoxicity, and it was successfully employed to image 4-month APP/PS1 mice in vivo. The in vitro fluorescence staining of Aβ oligomers on age-matched Tg mouse APP/PS1 has been also performed with a probe characterized by two electron-donating N,N-diethylaniline recognition groups bonded to a single-boron difluoride bridge azafluvene as the strong electron-withdrawing group [80]. Finally, a novel fluoro-substituted cyanine probe, F-SLOH, demonstrated a good Aβ oligomer selectivity with a high binding affinity. The selectivity towards the Aβ oligomers in the brain was ascertained by in vitro labelling on tissue sections and in vivo labelling through the systemic administration of F-SLOH in 7-month APP/PS1 double-Tg and APP/PS1/Tau triple-Tg mouse models [81]. However, to our knowledge, all the listed fluorescent probes have not been studied for their specificity for amyloids associated with specific diseases. Due to their significant aromatic character, it is likely that these types of molecules are not able to discriminate between different amyloid proteins, thus preventing their employment as clinical tools for establishing the early and accurate diagnosis of neurodegeneration in different, but closely related, diseases.

### 3.2. Tau Probes

Tau is a highly conserved and soluble protein, classified among the so-called “Intrinsically Disordered Proteins” [82]. Six different Tau isoforms of 352–441 amino acids are currently known, which differ for the presence of zero (i.e., 0 N), one (i.e., 1 N), or two (i.e., 2 N) amino acid (AA) sequence inserts at the N terminal side of the protein, and by three or four AA small-sequence repeats at the C-terminus (commonly known as 3R and 4R, respectively); each isoform derives by the alternative splicing of the MAPT (microtubule-associated protein tau) gene [83,84]. Tau is mainly present in neuronal cells, where it is involved in the regulation of the stability of axonal microtubules and in the control of several other cellular signaling processes [85]. Upon a series of genetic and post-translational modifications (PTMs), Tau can present a reduced affinity for microtubules, resulting in their destabilization [85,86,87,88]. Moreover, when detached by microtubules, Tau can aberrantly accumulate into the cell cytoplasm, aggregating into toxic multimeric complexes responsible for neuronal cell death [89,90]. Indeed, several clinical and research findings have shown in recent years that aberrant aggregates of Tau participate in the development and progression of a number of neurodegenerative disorders and dementias, collectively named as tauopathies [54,91]. As a consequence, Tau has become a relevant therapeutic target for the development of agents disrupting aberrant aggregates or preventing their formation. However, no drugs have been approved to date for the treatment of tauopathies. Furthermore, Tau PTMs have emerged as potential biomarkers for the early identification and diagnosis of tauopathies, and research endeavors have also been conducted towards their identification in human biofluids, especially by the means of non-invasive techniques [14,92,93,94,95,96,97,98]. Imaging and detection in Tau aggregates is of primary importance, and significant research has also been devoted towards the identification of probes that facilitate the monitoring of different tauopathies; the diagnosis of such diseases is still based on imaging techniques and clinical evaluation is often confirmed only after an examination of patients’ brains [99,100,101,102]. Several methods are currently available to help in this respect, with positron emission tomography being one of the most employed for neuroimaging deposits of Tau. Indeed, PET presents several advantages for the diagnostic imaging of Tau, including its relatively low invasiveness and a number of already reported molecular probes targeting this protein with a good specificity and affinity, also in vivo [103,104]. Such probes can help to detect abnormal Tau aggregates accumulating in different districts of the human brain, already at the early phases of neurodegeneration. Hence, they represent valuable complementary tools for monitoring tauopathies’ progression [103]. Moreover, PET imaging might help to differentiate between tauopathies based on different Tau isoforms (e.g., 3R, 4R, and 3R + 4R), through the use of specific tracers [105]. A number of PET probes are currently available for detecting Tau deposits in preclinical and clinical settings (Figure 3 and Table 1). One of the first reported Tau tracers is [^18^F]-FDDNP, which has demonstrated valuable performances in AD monitoring, albeit showing a poor ability to differentiate aggregates of Tau from Aβ-related ones [106]. In addition, [^11^C]-PBB3 [107] has demonstrated being able to differentiate AD patients from healthy controls, and is able to detect Tau aggregates in subjects with dementias not related to Alzheimer’s disease; however, this compound poorly discriminated among Tau/Aβ deposits [108]. Tau tracers based on a quinoline scaffold, which are also known as “THK compounds”, have also been reported [109,110,111,112,113,114,115,116], demonstrating a significant selectivity for Tau aggregates with respect to Aβ-related ones. Among the first of this class are [^18^F]-THK-523, [^18^F]-THK-5105, [^18^F]-THK-5116, and [^18^F]-THK-5117. [^18^F]-THK-523 has also displayed low accumulation in patients’ brains, while [^18^F]-THK-5105 and [^18^F]-THK-5117 showed uptakes well correlated with Tau-related disease progression [109,110,111,112,113,114,115].

Later, [^18^F]-THK5351 demonstrated improved PET imaging performances [116]. However, it also showed issues related to poor absorption in patients’ brains, concurrently with the administration of monoamine oxidases (MAOs) inhibitors; this is an issue that has been highlighted for several probes reported for Tau PET imaging [116]. Additionally, compounds based on the 5,9b-dihydro-4aH-pyrid[4,3-b]indole scaffold have also been investigated, with Flortaucipir (18F) (i.e., [^18^F]-AV-1451, [^18^F]-T807) being the only radiotracer approved for imaging Tau deposits in vivo [117,118]. This molecule demonstrated being promising in clinical settings and showed no contraindications for Tau PET imaging; scenarios in which Flortaucipir (18F) can be used for diagnosis and monitoring have also been suggested [118,119], as well as its limitations in detecting early-stage Tau pathology [120]. Notably, Flortaucipir (18F) has shown a good BBB permeability and high affinity for Tau paired helical filament (PHF) aggregates in patients with Alzheimer’s disease [121]. However, 18F-flortaucipir can accumulate into pigment-containing and calcified structures in the brain, and it can also bind to monoamine oxidases, thus potentially affecting its specificity in imaging analyses [122,123,124].

Overall, several among the first reported PET probes of Tau present limitations, including, for example, a low selectivity for Tau versus Aβ aggregates in some cases, and a low specificity for white matter in the brain, thus altering imaging contrast [103,119,125]. Moreover, several of these probes were not able to discriminate between different Tau deposits. These limitations fueled the development of novel Tau PET tracers such as [^18^F]-GTP1, [^18^F]-JNJ069, [^18^F]-JNJ311, [^18^F]/[^3^H]-MK-6240, [^18^F]/[^3^H]-PI-2620, [^18^F]/[^3^H]-RO-948 (i.e., [^18^F]/[^3^H]-RO6958948), [^18^F]-APN-1607 (i.e., [^18^F]-PM-PBB3), [^11^C]-RO6931643, [^11^C]-RO6924963, and [^18^F]-SNFT-1 [126,127,128,129,130] (Figure 3), with these compounds showing significantly less off-target binding to aggregates and better pharmacokinetic properties. Moreover, several of these compounds showed improved pharmacokinetic properties and the ability to discriminate AD from non-AD tauopathies. Examples in this regard are the compounds [^18^F]-MK-6240, [^18^F]-PI-2620, [^18^F]-RO-948, and [^18^F]-APN-1607, which showed good performances in discriminating AD from non-AD patients in in vivo PET imaging studies [131,132,133,134], and also helping to detect low levels of Tau. Altogether, these data can offer insights into the design of compounds targeting Tau aggregates [135,136,137,138]. The utility of Tau PET tracers for the early diagnosis and monitoring of AD has also been studied in combination with assessments of phosphorylated Tau, providing remarkable results [139]. However, further research is needed to better assess their utility in clinical settings [103,119,131]. In particular, most of the studies reported so far focused on AD and related dementias, and, to a lesser extent, CBD (Corticobasal Degeneration), CTE (Chronic Traumatic Encephalopathy), FTLD (Frontotemporal Lobar Degeneration), PiD (Pick’s Disease), and PSP (Progressive Supranuclear Palsy). In this respect, future research focusing on additional tauopathies is expected to provide fruitful insights into a better understanding of Tau-related diseases. While PET tracing is an established method for neurodegenerative diagnosis, imaging pathological Tau might also present some intrinsic limitations. For example, results of in vivo Tau PET imaging performed in post mortem brains of AD patients often resulted in not being fully in agreement with disease progression [125]. In addition, the high costs and exposure risks associated with the use of radioligands cannot be neglected, as this could hamper routine monitoring. It is worth noting that studies on the use of benzimidazole compounds such as lansoprazole and astemizole (Figure 3) as potential PET tracers have also been conducted [140,141], demonstrating that they might have great potential in radio-labelled neuroimaging for the in vivo early detection of AD. To date, no successful stories focusing on the repositioning of already approved drugs towards Tau PET imaging have been reported. However, these aspects are of particular interest and could suggest novel sources and approaches for rapidly identifying PET tracers for tauopathies investigations. Besides PET, magnetic resonance imaging has also been used to provide insights into AD and tauopathies in different settings [141,142]. Indeed, MRI presents several advantages with respect to PET for Tau imaging. For example, it does not employ ionizing radiation in its assessments. While such techniques show large possibilities of application for studying changes in the brains of patients, approaches and probes based on MRI potentially useful for the molecular imaging of Tau are still under development [141,142,143,144,145]. Indeed, only few studies have reported the use of MRI-based Tau imaging with contrast agents, the majority of them being focused on animal models and employing fluorinated compounds [142]. One among the first studies reporting the use of a fluorine-19-labeling compound (i.e., Shiga-X34, Figure 3), by the means of fluorine-19 magnetic resonance imaging (19F-MRI) [146,147], was reported by Yanagisawa et al. in 2018 [148]. In that study, the authors performed ex vivo analyses on rTg4510 mice, demonstrating that their compound Shiga-X34 can bind to NFTs of different tauopathies. In this study, Shiga-X34 was also used as a starting point for the development of a 19F-MRI additional probe for Tau imaging (i.e., Shiga-X35). Notably, Shiga-X35 (Figure 3) showed efficient detection abilities for Tau NFTs, as observed by the means of 19F-MRI imaging in rTg4510 mice, and readily reached the brain after injection, although was gradually excreted with some undesirable accumulation [148]. However, Shiga-X35 presents a low specificity and selectivity for NFTs and senile plaques, suggesting that further optimization on this scaffold is required. Besides Shiga-X35, an additional MRI probe (i.e., the DNA-aptamer nanoparticle TauX) was reported by Badachhape et al. [149]. Notably, this product demonstrated enhancing the MRI signal in the transgenic PS19 mice line, 4 to 6 months before the subjects showed tauopathy symptoms. As a further note, studies employing artificial intelligence (AI) techniques to analyze MRI-image-related data and other readily available information for predicting the clinical status of patients with neurodegenerative diseases have very recently been proposed and applied for amyloids [150,151,152,153]. An example of this comes from a very recent study by Lew and colleagues [154], who reported the development and application of a deep learning approach able to predict the PET-determined amyloid-tau-neurodegeneration (ATN) biomarker status from MRI and other patient-related data reported in the Alzheimer’s Disease Neuroimaging Initiative (ADNI) database (www.adni-info.org, accessed on 29 November 2023). Notably, the study demonstrated good prediction performances and the agreement of the developed AI models with respect to the PET outcomes for ATN biomarking status. While the study presents some limitations, the results suggest that future advances in MRI techniques and their integration with AI approaches might provide diagnostic performances comparable to PET imaging, with reduced costs, time, and risks.

To the best of our knowledge, there are no PET or MRI probes for selectively imaging Tau soluble aggregated species as oligomers, though these are considered to be among the most toxic components in tauopathies. In this context, fluorescent-probing techniques might provide several advantages, such as improved cost effectiveness and a high sensitivity and specificity. Indeed, in the last years, fluorescent probes such as thioflavin T (ThT) or thioflavin S (ThS) (Figure 3), whose fluorescence increases upon binding to Tau aggregates, have been introduced in experimental assays assisting the design of preclinical candidates [155,156]. However, their applicability is restricted only to in vitro experimentations, and they very often require complementary experimental confirmations due to their low binding specificity towards different aggregates. Imaging Tau aggregates in vivo can be challenging due to their structural similarity to the structure of Aβ fibrils; however, fluorescent probes that specifically detect Tau aberrant deposits have been very recently studied [157,158,159,160]. Among them, we can find the quinoline-based fluorescent probes (i.e., Q-tau 1 to 4, Figure 3) reported by Elbatrawy et al. [157], which showed a high selectivity towards Tau aggregates in ex vivo samples from AD brain tissues. Indeed, compounds Q-tau1 and Q-tau4 showed significant binding affinities toward Tau deposits, with selectivity indices of 4.4 and 3.5 over Aβ aggregates, respectively. Moreover, Q-tau4 also showed: (i) a fluorescence signal that positively correlated with the immunofluorescence of p-tau; (ii) low cytotoxicity; and (iii) a selective fluorescent profile different from that of Aβ plaques in AD brain tissues. Furthermore, in 2020, Zhao and co-workers reported compound pTP-TFE (Figure 3) that selectively binds to soluble Tau aggregates over mature fibrils [161]. Of note, this compound was developed starting from the p-FTAA probe (Figure 3) reported in 2009 by Åslund and colleagues [162], which showed a good BBB permeability and different spectral signatures when bound to Aβ or Tau deposits; pTP-TFE was also demonstrated to be cell permeable. As a consequence, this compound represents a promising proof of concept tool for the study of tauopathies’ development and progression [161]. Later, in 2021, Oh et al. [163] developed two additional series of thiophene derivatives (Figure 3) showing an improved selectivity for Tau aggregates, also in in vitro experiments on SH-SY5Y cells stably expressing GFP-tagged Tau. More recently, in 2022, Soloperto et al. [164] reported a focused library of eight BODIPY-derived probes, i.e., from BT1 to BT8, inspired by the selective fluorescent probe TAU1. These compounds feature conjugated systems of 13–19 Å length, ending with an electron donor group and characterized by a different polarity. Of note, one of them (i.e., BT1, Figure 3) showed favorable photophysical properties and a high selectivity towards phosphorylated and oligomeric Tau in an in vitro humanized cellular model; the results reported in this study paved the way towards the optimization of compounds potentially aiding in the early diagnosis of neurodegenerative diseases.

While the development of fluorescent imaging Tau probes is of primary interest, this field of research is still unfortunately in its infancy. Indeed, most of the fluorescent probes discussed above have been reported only very recently and studied in in vitro assays or ex vivo samples. Therefore, their potential translation into clinical use has yet to be proven. However, considering their potential for the diagnosis and monitoring of tauopathies, future research on fluorescent-based probes of Tau is warranted.

### 3.3. α-Synuclein Probes

#### 3.3.1. Florescent Probes

In the following paragraph, the current development of fluorescent probes for detecting αSyn aggregation in ex vivo and in vivo systems is presented. A description regarding their selectivity against a specific amyloid or aggregate type is provided in Table 1.

Anle138b is a pyrazolo bearing 1,3-benzodioxole as a substituent in position 3 and 3-Br-phenyl in position 5 (Figure 4). Studies from Fields et al. show that it inhibits the formation and aggregation of αSyn oligomers, without impacting the protein expression and physiological function. Also, the authors demonstrated that it decreases oligomer accumulation, neuronal degeneration, and PD progression in mice [165]. Interestingly, the anti-aggregation activity of the compound can be correlated to its 1,3-benzodiaxole ring. In cells, the methylene bridge between the oxygens might be disrupted, thus leading to two free hydroxyl groups. Although this could be the basis for the interaction with αSyn, experimental data are needed to prove it. Notably, other experiments showed that its efficacy in disrupting oligomeric aggregates was due to the ability of intercalating among the β-sheets structures, present in the core of these species [165]. Finally, its efficacy was also proven against Tau and amyloid β aggregates [166]. In addition to inhibiting αSyn oligomerization, anle138b showed significant fluorescence enhancement at around 345 nm after binding to αSyn oligomers and fibrils. Interestingly, anle138b was able to interact and bind αSyn fibrils with a K_d_ value of 190 ± 120 nM. When the binding with αSyn fibrils took place, an anle138b fluorescence increase occurred around 335 nm and was correlated with the decrease in the red wing fluorescence at λ > 385 nm [167,168]. Due to the promising anti-aggregation activity of the compound, as well as its ability to bind αSyn aggregates and displaying a strong fluorescence shift upon binding, anle138b provided a new possibility for the early diagnosis of PD. In particular, anle138b was proposed as a probe for the in vivo detection of retinal αSyn deposition in combination with confocal scanning laser ophthalmoscopy as a non-invasive technique to spot early αSyn aggregation. However, some limitations have hampered its development as an αSyn fluorescent probe in non-invasive in vivo detection. The main limits are the moderate affinity for αSyn aggregates, together with the lack of affinity regarding amyloid species (αSyn, Tau, and amyloid β) and its spectral properties [168,169,170]. However, this study indicated that αSyn aggregation inhibitors might be a way to exploit the development of imaging probes.

Novel benzo-thiazole derivatives as potential fluorescent probes were developed by Watanabe et al. in 2017 [171]. Among them, PP-BTA-4 (Figure 4) presented an excitation/emission profile in chloroform of 559/727 nm. Upon binding with αSyn aggregates, PP-BTA-4 presented a considerable fluorescent shift to 682/782 nm. Furthermore, this compound showed a high binding affinity for αSyn aggregates in vitro with a K_d_ value of 48 ± 0.6 nM. As the background autofluorescence was low in the near-infrared region (650−900 nm), the emission wavelength of this compound was considered to be adequate for the detection of αSyn aggregates in brain sections [171]. In human PD brain slices, where Lewy bodies were present, PP-BTA-4 stained some regions that were positive for immunohistochemical staining with a specific αSyn antibody [171]. However, this compound lacked amyloid selectivity. In fact, it was able to recognize and bind to Aβ_1–42_ aggregates in vitro and in ex vivo samples (AD brain slices with Aβ_1–42_ plaques).

Although PP-BTA-4 had no selectivity for αSyn and Aβ aggregates, it contributed to filling the gap of near-infrared (NIR) fluorescence probes to detect αSyn fibrils. Through optimization and chemical modification, the compound may serve as an NIR fluorescent probe for the detection of αSyn and Aβ_1–42_ aggregates in human brains [171].

These studies elucidate the lack of a specific, highly selective fluorescent probe for the detection of αSyn inclusions in ex vivo and in vivo samples. The main limit characterizing the candidates developed up to now is the affinity for a determined αSyn species (oligomers and fibrils), as well as amyloid specificity (selectivity towards αSyn, Aβ_1–42_, or tau). Although some progress has been made (potential in vivo imaging of αSyn with the molecule anle138b), more research is needed to optimize these derivatives to propose a suitable and effective fluorescent probe. A way to improve the affinity may be to rationally design compounds able to bind to a specific portion of αSyn, preferably involved in the core of early-stage aggregates like the pre-NAC or NAC region [172,173]. In this way, a high selectivity and affinity may be expected. Furthermore, by detecting early-stage aggregation, the probe could be of crucial importance in the early diagnosis of neurodegenerative diseases.

#### 3.3.2. Radiolabeled Probes

There are a few main characteristics that a compound can bear to be identified as a suitable radiolabeled probe (radiotracer). First, a good affinity (nM scale) and high specificity for αSyn are required, since αSyn aggregates in the brain are way less concentrated than other amyloid inclusions (like Aβ plaques) [174]. Then, these molecules must display appropriate pharmacokinetics properties, such as a suitable lipophilicity (log P = 1−3) and molecular weight (<600 Da). In fact, these optimized parameters allow the compound to cross the BBB and neurons membrane to interact with intracellular αSyn deposits. Finally, as we saw for the fluorescent probes, they must display a low toxicity profile and an adequate brain clearance, together with chemical stability and a good brain uptake [174]. In the following paragraph, the current development of radiolabeled probes to detect αSyn aggregation in ex vivo and in vivo systems is presented (see Table 1 for selectivity against a specific amyloid or aggregate type).

Ten years ago, Bagchi et al. synthesized phenothiazine derivatives able to selectively bind to αSyn fibrils [175,176]. Among them, SIL23 (Figure 3) was the one displaying a moderate affinity for αSyn fibrils, but had a high selectivity versus Aβ and tau. By radio-synthesis, the researchers eventually converted the compound into [^125^I]SIL23 to create a radiolabeled probe. In vitro binding assays showed that [^125^I]SIL23 could bind to αSyn aggregates in human PD brain homogenates with a K_d_ value of 143 ± 24 nM. Also, the compound was able to recognize αSyn in transgenic PD mice (M83) with a K_d_ of 151 nM, but not in healthy controls. The main limit for the further development of this compound was its moderate affinity, which hampered the possibility of performing biodistribution and pharmacokinetic studies [175]. Among the phenothiazine derivatives, other interesting compounds were SIL5 and SIL26, which were also radiolabeled to obtain [^11^C]SIL5 (Figure 4) and [^18^F]SIL26 [176]. Ex vivo biodistribution studies in rats showed that both [^11^C]SIL5 and [^18^F]SIL26 (Figure 4) could penetrate the BBB and demonstrated appropriate washout kinetics. However, [^18^F]SIL26 was the compound showing the lowest brain uptake, namely a 0.758 ± 0.013% injected dose (ID)/g at 5 min after injection. Thus, the only compound suitable for further evaluation was [^11^C]SIL5. MicroPET imaging of [^11^C]SIL5 in a healthy macaque confirmed that it could cross the BBB and showed a homogeneous distribution. However, its moderate affinity towards αSyn fibrils binding, together with a modest brain uptake (0.953 ± 0.115%ID/g), hampered its further application in human PET imaging [175].

In 2015, Chu et al. designed and synthesized a series of 3-benzylidene-indolin-2-one analogues to develop αSyn PET radiotracers [177]. Among them, compound 46a bore a fluoroethyl side, which could be easily transformed in a ^18^F labeled tracer. This molecule, namely [^18^F]46a (Figure 4), displayed a high affinity for αSyn fibrils (K_d_ = 2.1 ± 0.3 nM) and good selectivity versus Aβ and Tau fibrils (K_d_ = 142.4 ± 36.9 nM and 80.1 ± 12 nM, respectively), as displayed by in vitro saturation binding assays [177]. Up to now, [^18^F]46a is one of the probes with the greatest affinity for αSyn fibrils [174]. However, when authors tested the capacity of the compound to stain αSyn fibrils in PD brain slices, they could not obtain a reliable quantification of the detected aggregates. This is probably due to the high log *p* value (4.18) of [^18^F]46a, which might cause nonspecific binding [177]. Also, further development of this potential probe was hampered by chemical instability issues. In fact, the nitro group might be reduced to the amino group in living systems. Consequently, [^18^F]46a was not suitable for serving as a PET radiotracer [177].

Radiolabeled diphenyl derivatives were synthesized a few years ago by Ono et al., with the aim of developing αSyn aggregates ligands [178]. Among them, [^125^I]IDP-4 (Figure 4) displayed the highest affinity for αSyn aggregates at K_d_ values of 5.4 ± 1.5 nM with a 3-fold selectivity versus Aβ aggregates. Another good hit was [^125^I]IDP-3, which was able to bind to αSyn with a K_d_ value of 23.3 ± 2.8 nM, as displayed by in vitro binding saturation assays [178]. Notably, fluorescent staining of PD brain sections showed that both compounds could bind to αSyn aggregates in LBs. However, ex vivo biodistribution in mice showed that [^125^I]IDP-4 and [^125^I]IDP-3 (Figure 4) were characterized by a low brain intake. Therefore, neither compound was suitable for in vivo αSyn imaging [178]. Nevertheless, their capacity to bind to LBs αSyn aggregates opened the possibility of further chemical modifications on the diphenyl core, which may lead to an increased brain uptake and possible development as a PET imaging probe.

More recently, Verdurand et al. synthesized the benzoxazoles derivatives 2FBox and 4FBox to selectively recognize αSyn fibrils [179]. The compounds were then radiolabeled with ^18^F to assess their suitability as radiotracers. As displayed by in vitro saturation filter binding assays, [^18^F]2FBox (Figure 4) displayed the highest affinity for αSyn recombinant fibrils (K_d_ = 3.3 ± 2.8 nM). Since the binding assays were promising, the authors decided to further evaluate the compounds using in vitro autoradiography with αSyn- and Aβ_1−42_ fibril-injected rats, a transgenic PD mice model (M83), and a transgenic AD mice model (PDAPP line J20) [179]. Both compounds could detect Aβ and αSyn fibrils in a non-selective manner in the fibrils-injected rats, but failed to recognize Aβ_1–42_ and αSyn aggregates in the other mice models [174,179]. To obtain more insights on the activity of those candidates towards amyloid aggregates’ detection, the authors carried out imaging experiments on post mortem brain sections (PD, MSA, and AD patients). Surprisingly, neither [^18^F]2FBox or [^18^F]4FBox (Table 1) could detect Aβ_1–42_ and αSyn aggregates, while in vivo PET imaging of rats showed that the compounds could cross the BBB with a good initial brain uptake but failed to detect amyloids fibrils in the fibril-injected rats. Due to these limitations, namely amyloid non-selectivity and a lack of aggregate detection ex vivo and in vivo, [^18^F]2FBox and [^18^F]4FBox were not suitable as PET radiotracers for αSyn [179].

Anle253b is a diphenyl pyrazole sharing a similar structure with that of anle138b [168], identified by Maurer et al. as a possible PET tracer. Bearing an accessible methyl group, the compound was suitable for ^11^C methylation; thus, it was selected and screened as a PET imaging probe. Interestingly, anle253b bound to αSyn fibrils with an IC_50_ value of 1.6 nM. This very good affinity profile pushed the authors to perform in vivo PET studies in healthy rats [180]. These experiments showed that [^11^C]anle253b (Figure 4) could cross the BBB with medium brain uptake and was distributed homogeneously in the rat brain. However, data regarding its uptake dynamics, as well as selectivity versus other amyloid aggregates, have yet to be collected and optimized. In particular, the compound displayed a high lipophilicity (logP = 5.21), which may be responsible for the poor pharmacokinetics of [^11^C]anle253b in vivo. Thus, although the preliminary data regarding the brain uptake and bioavailability of [^11^C]anle253b are promising, at the current stage, it cannot be employed as a PET tracer in humans [174,181]. The optimization of anle253b, in particular the replacement of a phenyl ring with pyridine to decrease the compound’s lipophilicity, led to MODAG-001 [182]. To make the compound suitable for PET imaging scanning, Kuebler at al. radiolabeled MODAG-001 with ^3^H and ^11^C [182] (Table 1). Thus, the researchers carried out a series of in vitro and in vivo biological evaluation studies with both [^11^C]MODAG-001 and [^3^H]MODAG-001. In particular, [^3^H]MODAG-001 displayed an impressive high binding affinity for αSyn fibrils in vitro, with a Kd value of 0.6 nM. Furthermore, it showed a 30-fold selectivity over Tau and Aβ_1–42_ aggregates [181]. Additionally, [^11^C]MODAG-001 was able to efficiently penetrate the mouse brain (SUV = 1.4). Interestingly, thanks to metabolic studies, the authors noticed that one of the three main metabolites of [^11^C]MODAG-001 was represented by the demethylated derivative. Thus, being afraid that demethylation might hamper PET quantification imaging, the researchers fully deuterated the nonradioactive methyl group and synthesized (d3)-[^11^C]MODAG-001. By in vivo PET imaging, the authors evaluated the binding properties of (d3)-[^11^C]MODAG-001 in three αSyn-fibril-inoculated rats and one noninjected control. Interestingly, significantly higher signals were observed in the three αSyn-fibril-inoculated rats at 4 days after (d3)-[^11^C]MODAG-001 injection [182]. This was indicative of the compound’s good binding profile towards αSyn aggregates. To further characterize (d3)-[^11^C]MODAG-001 as a possible PET tracer, the researchers carried out in vitro autoradiography in LBD (Lewy Bodies Dementia), PSP (Progressive Supranuclear Palsy), AD, and healthy control cases. Surprisingly, no strong binding was observed in any of the LBD brain sections, but Aβ plaques were observed in the AD brain tissues [174,182]. Furthermore, no Tau fibrils were shown in the PSP and AD cases. This can be explained in terms of non-specific binding and low target availability in LBD brains.

Recently, Kaide et al. designed and synthesized bis-quinoline derivatives as useful probes for in vivo αSyn imaging [183]. To assess the activity of these derivatives in detecting αSyn aggregates, the authors first performed competitive inhibition assays in vitro. Notably, two compounds (BQ-1 and BQ-2) showed a high affinity towards αSyn fibrils (K_i_ equal to 17.0 and 11.6 nM, respectively) [183]. Interestingly, fluorescent staining experiments proved the efficacy of BQ-1 and BQ-2 in detecting αSyn aggregates in PD human brain samples. Since BQ-2 showed the highest affinity towards αSyn aggregates, it was chosen for radiolabeling and further evaluations. Thus, the authors evaluated the lipophilicity of [^18^F]BQ-2 (Figure 4) and performed biodistribution studies in vivo (mice). Interestingly, the compound displayed a moderate initial brain uptake (1.59%ID/g at 2 min postinjection) and was characterized by a moderate log D (2.62) [183]. Despite its promising pharmacokinetic profile, [^18^F]BQ-2 was primarily retained in the brain (1.35%ID/g at 60 min postinjection), probably because of nonspecific binding. This disadvantage was confirmed by in vitro autoradiography. In fact, a large amount of nonspecific binding was observed in the whole brain section [183]. Finally, this suggests that [^18^F]BQ-2 was not a suitable PET probe for imaging αSyn aggregates in PD patients’ brains.

In comparison with fluorescent probes developed for the imaging of αSyn in ex vivo and in vivo samples, the research on radiolabeled probes has taken several steps forward. In fact, [^125^I]IDP-3 and [^125^I]IDP-4 were able to stain αSyn in LB ex vivo samples, while other candidates like [^3^H]MODAG-001 and [^18^F]46a were characterized by a potent affinity for αSyn fibrils [177,178,182]. Furthermore, most of the compounds discussed in this paragraph were able to cross the BBB and spread homogenously in the brains of in vivo models. However, no suitable radiolabeled probe has yet been obtained for αSyn imaging. The main limits are represented by a low brain uptake, low amyloid selectivity, chemical instability, and low affinity for specific αSyn species [174]. As a result, more efforts are required to develop a suitable compound able to be exploited as an αSyn imaging probe [180].

## 4. Conclusions

The lack of early diagnoses of neurodegenerative diseases (e.g., AD and PD) underscores the critical need for inexpensive, easy-to-access probes able to detect the early onset of the pathologies. While relevant variations in biomarkers of specific proteinopathies have been identified even several years before the onset of the disease, to date, the clinical diagnosis of most of neurodegenerative disease relies on PET or MRI imaging. However, the application of such techniques presents several limitations. These include the high costs related to the analyses, the risks to health for patients that are continuously under monitoring, and, in some cases, the low accuracy in discriminating among the different neurodegenerative diseases. In addition, only post mortem analyses performed on patients’ brain tissues are able to correctly discriminate between different pathologies [4]. Furthermore, current diagnostic methods employing imaging technologies and biofluid analysis still encounter challenges, often related to low detectability and invasiveness issues. However, much progress has been made in recent years regarding Aβ_1–42_ imaging, in particular towards the detection of early toxic aggregation species such as oligomers, for example through the identification of novel and more effective probes. Examples in this respect come from the fluorescent, aromatic BD-Oligo and PTO-29 probes, which showed good photophysical properties, a high selectivity towards proteins oligomers, good blood–brain barrier penetration, low cytotoxicity, and successful in vivo imaging in mouse models [77,79]. Additionally, the fluoro-cyanine probe F-SLOH demonstrated a good Aβ oligomer selectivity with a high binding affinity ex vivo and in vivo [81]. However, data are still missing regarding their selectivity towards different amyloid proteins and neurodegenerative diseases. Similar considerations arise also for Tau species. Indeed, significant advancements have been observed in both biomarkers monitoring and in the field of PET and fluorescent probe imaging in recent years. With regard to biomarkers monitoring, most of the progress observed so far has come from the introduction of techniques enabling the identification of Tau species in large populations of patients, in non-invasive fluids, and the simultaneous monitoring of multiple Tau-related biomarkers. In addition, the discovery of PET tracers such as [^18^F]-MK-6240, [^18^F]-PI-2620, [^18^F]-RO-948, and [^18^F]-APN-1607 has enabled the discrimination of AD patients from controls, as displayed by in vivo studies [131,132,133,134]. Similarly, fluorescent probes such as Q-Tau 1 to 4, pTP-TFE and BT1 showed a low toxicity and high affinity for Tau aggregates towards Aβ_1–42_ [157,161,164]. However, it should be pointed out that the above-mentioned PET probes displayed low brain permeability, and the latter fluorescent ones were tested only in ex vivo samples. Thus, their development for human in vivo imaging has yet to be challenged. Finally, regarding αSyn, fluorescent probes able to detect early-stage aggregates ex vivo or in vivo are still in their infancy and require a drastic optimization. In fact, only Anle138b showed good results in detecting oligomers and fibrils in vivo, but its lack of selectivity towards different amyloid species stopped it from further development [168]. Regarding radiolabeled probes, some progress has been made in the PET imaging field. For example, the probe [^11^C]SIL5 showed good brain penetration and favorable pharmacokinetics properties, but had a modest selectivity for αSyn high-ordered aggregates (fibrils) [175]. On the contrary, compounds BQ-1 and BQ-2 displayed a high selectivity for αSyn fibrils, but a low brain uptake [183]. Again, these limitations hampered the development of these molecules for the early diagnosis of neurodegenerative diseases [180].

Even if the advent of near-infrared fluorescent probes presents a promising avenue for the non-invasive visualization of amyloid aggregates, challenges in specificity, selectivity, and clinical translation persist, urging further exploration into new chemical biology tools and probes. In this review, we offered a nuanced assessment of the achievements, drawbacks, and potential directions in the development of fluorescent and radiolabelled probes for neurodegenerative disease diagnosis. By critically evaluating the current state of the art, this review aims to propel advancements in neuropathology, steering toward preventive measures and early diagnosis with enhanced precision and accessibility in the realm of neurodegenerative diseases.

## Figures and Tables

**Figure 1 molecules-29-00722-f001:**
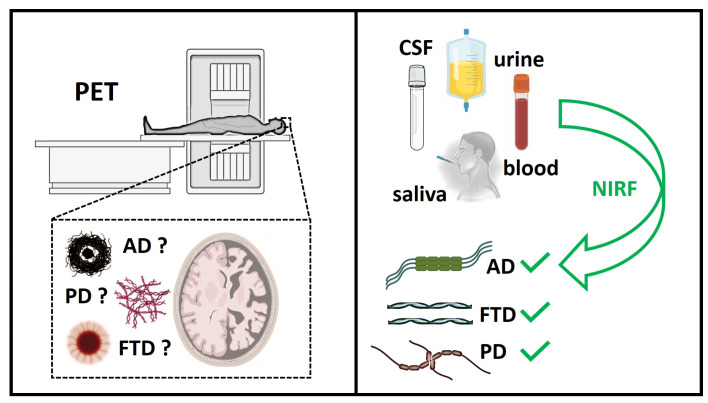
Schematic representation of two different diagnostic approaches in neurodegeneration through PET (Positron Emission Tomography) and Near-Infrared (NIR) fluorescent imaging (NIRF), discussed in this review.

**Figure 2 molecules-29-00722-f002:**
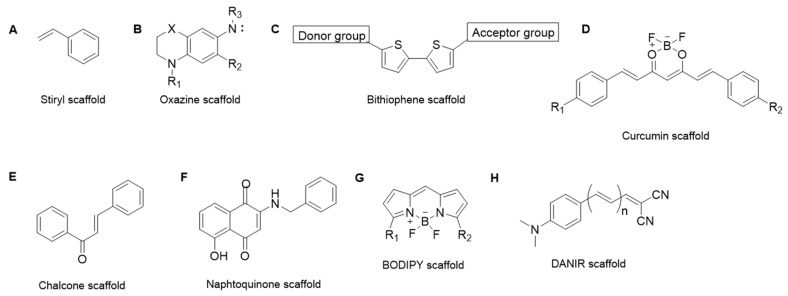
Representation of the different molecular scaffolds employed for the design of NIR probes detecting Aβ aggregates: (**A**) (Styril scaffold), (**B**) (Oxazine scaffold), (**C**) (Biothiophene scaffold), (**D**) (Curcumin scaffold), (**E**) (Chalcone scaffold), (**F**) (Naphtoquinone scaffold), (**G**) (Bodipy scaffold), (**H**) (DANIR scaffold).

**Figure 3 molecules-29-00722-f003:**
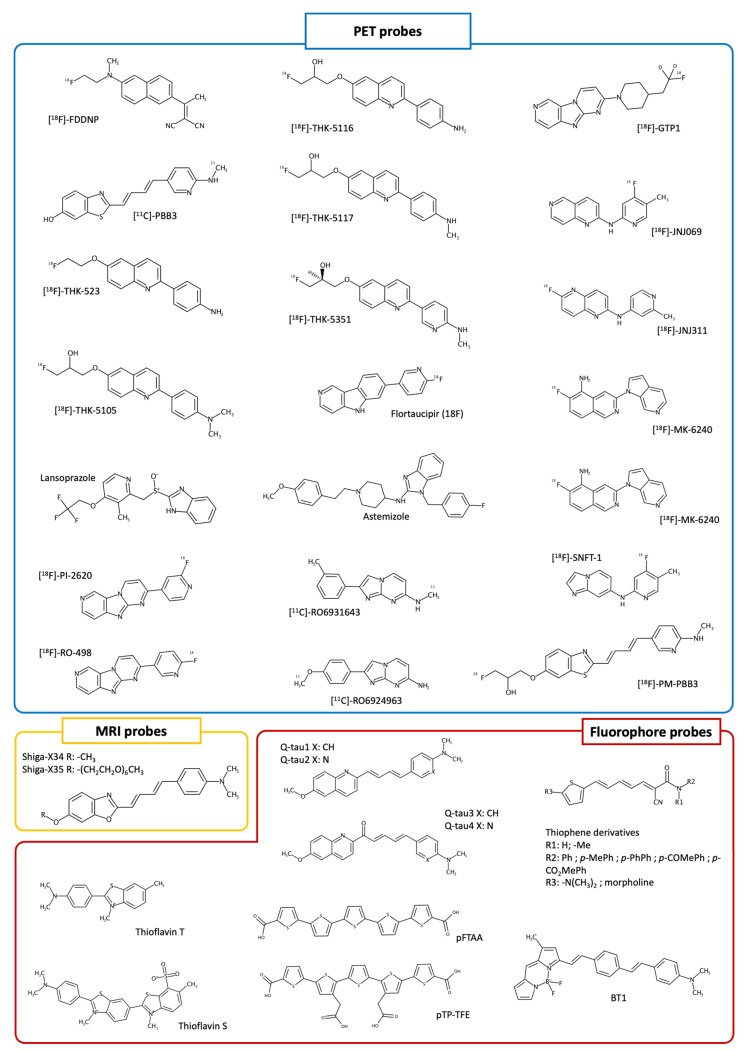
Chemical structure of PET, MRI, and fluorophore probes detecting Tau, discussed in the review.

**Figure 4 molecules-29-00722-f004:**
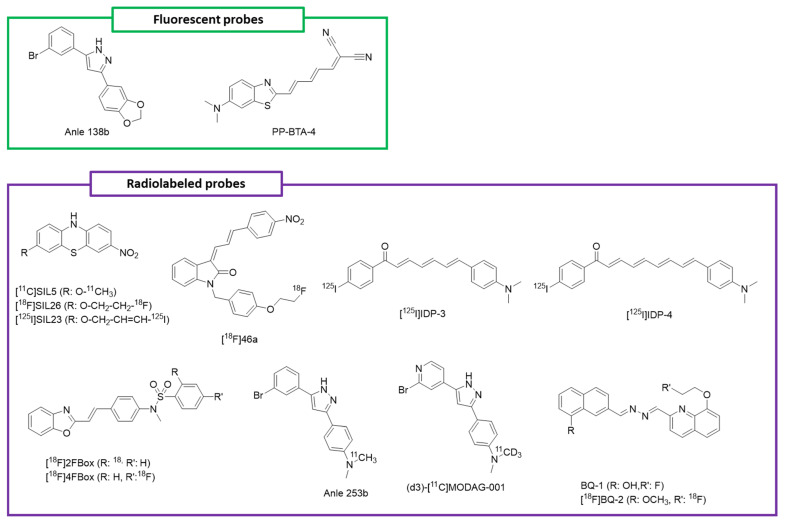
Probes for the detection of αSyn aggregation discussed in the review.

**Table 1 molecules-29-00722-t001:** In the table, the probes selectively recognizing one specific kind of amyloid aggregate are reported. The selectivity towards a specific amyloid species (αSyn, Aβ_1–42_, Tau) is also indicated, as well as the mechanism by which the probe interacts with its target.

Compounds	Aggregate Selectivity	Amyloid Selectivity	Mechanism of Interaction
Styryl derivatives	None (Aβ_1–42_ aggregates)	Aβ_1–42_	Possible intercalation in β-sheet-rich structures
Oxazine dyes	None (Aβ_1–42_ aggregates)	Aβ_1–42_	Possible intercalation in β-sheet-rich structures
2,2′-bithiophene derivatives	None (Aβ_1–42_ fibrils, amyloid aggregates)	None	Binding to amyloid fibrils surface
Curcumin derivatives	Oligomers and fibrils	None (αSyn, Aβ_1–42_, Tau)	Not specified
Chalcone derivatives	Aβ_1–42_ plaques	Aβ_1–42_	Not specified
1,4-napthoquinones	None (Aβ_1–42_ aggregates and plaques)	Aβ_1–42_	Not specified
BODIPY dies	Oligomers and fibrils	Aβ_1–42_	Binding to hydrophobic surfaces
CRANAD-3	None (different oligomers, monomers)	Aβ_1–42_	Not specified
BD-Oligo probe	Oligomers	Aβ_1–42_	Similar to BODIPY dies
BODIPY-6 and aza-BODIPY	Oligomers and fibrils	Aβ_1–42_	Similar to BODIPY dies
PTO-29	Oligomers	Aβ_1–42_	Not specified
F-SLOH	Oligomers	Aβ_1–42_	Not specified
[^18^F]-FDDNP	None (Tau aggregates)	Aβ_1–42_, Tau	Not specified
[^11^C]-PBB3	None (Tau aggregates)	Aβ_1–42_, Tau	Not specified
THK compounds ([^18^F]-THK-523, [^18^F]-THK-5105, [^18^F]-THK-5116, [^18^F]-THK-5117 and [^18^F]-THK5351)	None (Tau aggregates)	Tau	Not specified
Flortaucipir (18F)	Tau fibrils (PHF)	Tau	Not specified
[^18^F]-GTP1	Neurofibrillary tangles (NFT)	Tau	Not specified
[^18^F]-JNJ069	Neurofibrillary tangles (NFT)	Tau	Not specified
[^18^F]-JNJ311	Neurofibrillary tangles (NFT)	Tau	Not specified
[^18^F]/[^3^H]-MK-6240	Neurofibrillary tangles (NFT)	Tau	Not specified
[^18^F]/[^3^H]-PI-2620	Neurofibrillary tangles (NFT)	Tau	Not specified
[^18^F]/[^3^H]-RO-948	Neurofibrillary tangles (NFT)	Tau	Not specified
Shiga-X34	Tau NFT	Tau	Not specified
Shiga-X35	None (Tau NFT and other aggregates)	Tau	Not specified
Quinoline-based probes (Q-tau1 and Q-tau4)	None (Tau aggregates)	Tau	Not specified
pTP-TFE	None (Tau soluble aggregates)	Tau	Not specified
BODIPY-derived probes (BT1)	Tau oligomers (phosphorylated)	Tau	Similar to BODIPY dies
Anle138b	Oligomers and fibrils	None (αSyn, Aβ_1–42_, Tau)	Possible methyl bridge disruption leading to free hydroxyl (H-bond interaction), intercalation in β-sheet-rich structures
PP-BTA-4	Fibrils	αSyn, Aβ_1–42_	Not specified
[^11^C]SIL5	Fibrils	αSyn	Not specified
[^125^I]SIL23	Fibrils	αSyn	Not specified
[^18^F]SIL26	Fibrils	αSyn	Not specified
[^18^F]46a	Fibrils	αSyn	Not specified
[^125^I] IDP-3	None (αSyn aggregates in LBs)	αSyn	Not specified
[^125^I] IDP-4	None (αSyn aggregates in LBs)	αSyn	Not specified
[^18^F] 2FBox	None	αSyn and Aβ_1–42_	Not specified
[^18^F] 4FBox	None	αSyn and Aβ_1–42_	Not specified
Anle253b	Fibrils	αSyn	Not specified, probably similar to Anle138b
(d3)-[^11^C]MODAG-001	Fibrils	αSyn	Not specified
BQ-1	None	αSyn	Not specified
[^18^F]BQ-2	None	αSyn	Not specified

## Data Availability

Not applicable.

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
