# Peer review of "Early Diagnosis of Neurodegenerative Diseases: What Has Been Undertaken to Promote the Transition from PET to Fluorescence Tracers"

_molecules, 2024, doi:10.3390/molecules29030722_

Round 1

Reviewer 1 Report

Comments and Suggestions for Authors

The manuscrupt by Bisi et al.reviews the literature on the use of chemical probes interacting with different beta-sheet amyloid species derived from the aggregation of amyloid beta-1-42 peptide, tau and alpha-synuclein. These chemical probes are able to interact with these peptides at different stages of their aggregation thus forming species which can be detected by imaging or fluorescence techniques. While the interaction of these chemical probes with the different proteins is well documented in terms of activity and detection, and this review gives an interesting overlook on the different methodologies and sensitivity, the main problem resides on the selectivity of the probes for the different amyloid structures. This selectivity, being of utmost importance for the correct clinical identification of the neurogenerative diseases, is in many cases cited and described, but unfortunately not understood or at least, its origin is not reported. A description of the interaction of the amyloid aggregates with the chemical probes at the molecular level as well as the origin of their selectivity, would be extremely useful to the reader.

Apart from this the manuscript is well written with only few minor amendments needed:

1) Page 3 line 143: heteroneity instead of heterogenicity

2) Page 13 line 556: add “(Figure 3)” alfter “SIL23”

3) Check that all acronyms are defined somewhere in the text (e.g. QY presumably meaning Quantum Yield)

Comments on the Quality of English Language

Well written with only minor typos or amendments needed

Author Response

We thank the reviewer for the comments and suggestions. We agree on the relevance of this output, and we added a supplementary table (Supplementary Materials, Supplementary Table S1) where the probes having a determined selectivity for one specific aggregate species or amyloid are reported. Regarding the interaction at the molecular level, unfortunately literature is not exhaustive: most of the mechanism proposed are speculative, and not specifically described. However, we tried to report in the table what we found in this regard, for sake of exhaustiveness.

1) Page 3 line 145: heteroneity instead of heterogenicity

2) Page 13 line 556: add “(Figure 3)” alfter “SIL23”

The suggested corrections have been integrated in the text

3) All acronyms have been checked and detailed.

Reviewer 2 Report

Comments and Suggestions for Authors

The manuscript entitled “Early diagnosis of neurodegenerative diseases: what has been done to promote the transition from PET to fluorescence tracers” by Bisi et al., has summarized the development of chemical compounds for detecting amyloid aggregates of amyloid beta, tau and alpha-synuclein. The review further delineates the discoveries of novel agents which has specificity towards detecting the early stage oligomers, potent blood brain barrier penetration and low cytotoxicity. Overall this is a very well written review and can be published with some minor changes.

1.       It would be better if the authors include a section regarding the binding modes of these different classes of compounds with amyloid aggregates.  

2.       It would be easier if the authors provide a table classifying the compounds that selectively binds a specific amyloid aggregate for example oligomer, prefibrils and mature fibrils.  

Author Response

We thank the reviewer for the comments and suggestions. Regarding these points, as reported above, we implemented the paper with a table reporting the selective probes interacting with one particular type of aggregate and amyloid (see Supplementary Material, Table S1). When possible, we also reported the binding mechanism at a molecular level. 
